

# Dopamine receptor activation elicits a possible stress-related coping behavior in a wild-caught songbird

Melanie R. Florkowski[1] and Jessica L. Yorzinski[1,2]

[1] Ecology and Evolutionary Biology Program, Texas A&M University, College Station, TX, United States
[2] Department of Ecology and Conservation Biology, Texas A&M University, College Station, TX, United States

## ABSTRACT

Animals experience stress throughout their lives and exhibit both physiological and behavioral responses to cope with it. The stress response can become harmful when prolonged and increasing evidence suggests that dopamine plays a critical role in extinguishing the stress response. In particular, activation of the D2 dopamine receptor reduces glucocorticoids and increases coping behavior, *i.e.*, behavioral responses to adverse stimuli that reduce the harmful effects of stress. However, few studies have examined the effects of dopamine on the stress responses of wild species. We therefore tested the hypothesis that activation of the D2 dopamine receptor influences coping-like behavior in a wild-caught species. We recorded behavior of house sparrows (*Passer domesticus*) before and after they received injections of D2 dopamine agonists, D2 dopamine antagonists, or saline. House sparrows are common in urban environments and understanding how they cope with stress may help us better understand how animals cope with urban stressors. We found that the birds significantly increased biting of inanimate objects after the agonist but there was no change following the antagonist or saline. The biting of inanimate objects may be a mechanism of behavioral coping. This change in biting behavior was not correlated with general movement. This study supports the hypothesis that D2 dopamine receptor activation is involved in the regulation of the stress response in a wild bird.

## INTRODUCTION

Animals experience stress – a real or perceived threat to homeostasis – throughout their lives. This stress has the potential to negatively impact their survival and reproduction (*Chrousos, 2009*). Stressors such as predator encounters, conflicts with social competitors, and harsh weather can threaten survival through a reduction in foraging success and a weakening of the immune system, which can potentially even result in death (*Brown & Kotler, 2004*; *Lange & Leimar, 2004*; *Pravosudov et al., 2001*; *Cirule et al., 2012*). The presence of stressors can also suppress reproduction and result in the abandonment of

Corresponding author
Melanie R. Florkowski,
mflorkow@tamu.edu

dependent offspring (*Love et al., 2004*). Because stress can significantly reduce fitness, mechanisms have evolved to physiologically and behaviorally cope with it.

When a stressor is perceived, it triggers the activation of the hypothalamic-pituitary-adrenal (HPA) axis, beginning a hormonal cascade that ultimately releases glucocorticoids into the bloodstream (*Sapolsky, Krey & McEwen, 1986*). Elevated glucocorticoids, such as cortisol and corticosterone, disrupt normal functions and shifts the individual into an 'emergency life history stage' (*Wingfield et al., 1998*). This stage can temporarily suppress the immune system (*Shini et al., 2010*), mobilize energy stores, and modify behavior to prioritize survival. Behavioral modifications in this stage can include increasing anti-predator behaviors (*Thaker, Lima & Hews, 2009*), decreasing parental care, and increasing group coordination (*Raulo & Dantzer, 2018*).

Although physiological and behavioral stress responses are often beneficial during stress, they can become harmful when prolonged. This can occur when either the stressor is chronic or because the stress response persists after the stressor is gone. Prolonged activation of the stress response can cause physiological problems (*e.g.*, metabolic dysfunction and impaired reproduction (*Wingfield & Sapolsky, 2003*; *López et al., 2018*)) as well as cognitive issues (*e.g.*, impaired memory and decision making (*Voellmy et al., 2014*; *Aisa et al., 2007*)). The ability to efficiently extinguish the stress response is important for resilience against the negative physiological and behavioral effects of prolonged stress (*Romero & Wikelski, 2010*; *Vitousek et al., 2019*; *Zimmer et al., 2019*).

Growing evidence indicates that dopamine plays an important role in extinguishing the stress response (*Cabib & Puglisi-Allegra, 2012*; *Sullivan & Dufresne, 2006*). Dopamine concentration increased in the nucleus accumbens, striatum, and medial frontal cortex of rodents exposed to a physiological stressor (*Abercrombie et al., 1989*). Activation of one dopamine receptor in particular, the D2 receptor, has been shown to increase with stress and may mediate coping behaviors (*Cabib & Puglisi-Allegra, 2012*; *Lattin et al., 2019*), which are behavioral responses to adverse stimuli that reduce the harmful physiological effects of stress (*Schouten & Wiepkema, 1991*). Examples of coping behaviors in some mammals and birds include grooming, freezing, and biting at inanimate objects (*Giorgi et al., 2003*; *Henson et al., 2012*; *Reis-Silva et al., 2019*; *Koolhaas et al., 1999*; *Hori, Yuyama & Tamura, 2004*; *Savory, Seawright & Watson, 1992*). Coping behaviors in response to stress can be crucial to regulating arousal levels and maintaining homeostasis by regulating glucocorticoids following a disturbance such as a stressor. Some animals that exhibit coping behaviors have reduced glucocorticoids when they are exposed to a stressor (*Sato et al., 2010*; *Kostal, Savory & Hughes, 1992*). In addition to altering stress related hormones, coping behaviors also impact other markers of stress including reducing the occurrence of stomach lesions (*Koolhaas et al., 1999*). Some of these coping behaviors are dependent on dopamine in the mesostriatal dopamine region in the brain (*Jones, Mittleman & Robbins, 1989*) and blocking D2 dopamine receptors during stress results in elevated post-stress glucocorticoids and increased stress-related health problems (*Sullivan & Dufresne, 2006*; *Puri et al., 1994*; *Sullivan & Szechtman, 1995*).

Our current understanding of dopamine's role in animals' behavioral response to stress largely comes from studies on domesticated and laboratory animals (*Baik, 2020*).

The effects of stress have been shown to vary between wild and domestic animals, even in the same species, therefore it may not be possible to generalize dopamine's role in stress (*Cabezas et al., 2013*). There have so far been few studies investigating dopamine's effect on behavioral responses to stress in wild species. A study in wild-caught house sparrows (*Passer domesticus*) (*Lattin et al., 2019*) however found that increasing D2 receptor activation increased preening, a possible stress-related coping behavior (*Henson et al., 2012*). House sparrows are invasive in many areas of the world and are urban specialists that experience a wide range of stressors as a result of their proximity to humans (*Beaugeard et al., 2018*). Understanding their stress responses may lead to better understanding of how animals cope with urban stressors as well as this species' success in non-native urban environments.

The aim of this study was therefore to test the hypothesis that activation of the D2 dopamine receptor influences stress-related coping behaviors in a wild species. We tested this hypothesis using wild-caught house sparrows, an urban specialist songbird species that is a model system for studying bird behavior and physiology (*Hanson et al., 2020*). Similar to the impact of D2 activation on the coping behavior of laboratory and domesticated animals (*Sullivan & Dufresne, 2006*; *Puri et al., 1994*; *Cheng, Singleton & Muir, 2003*; *Dennis, Muir & Cheng, 2006*), we predicted that increasing D2 receptor activation would increase coping behavior and blocking D2 receptor activation would decrease coping behavior. To test these predictions, we peripherally administered selective D2 dopamine agonists and antagonists to manipulate the activation of dopamine receptors in the brains of house sparrows.

## MATERIALS AND METHODS

### Animals

Twenty adult male house sparrows were captured with baited traps between February and May of 2019 in College Station, TX. Due to logistical issues associated with catching female birds, we restricted this study to males only. Our sample size was similar to that used in a previous study (*Balthazart, Castagna & Ball, 1997*). The birds were housed in randomly-assigned pairs in cages ($0.6 \times 0.33 \times 0.3$ m) at Texas A&M University ($30°36'$ N, $96°21'$W) in an indoor room ('housing room': $5 \times 6.3$ m). They were kept on a 13 h:11 h light:dark cycle at $24.0 \pm 0.5$ °C (mean $\pm$ SD). They were individually marked with metal and colored leg bands. The pairs were given at least 7 days to acclimate to captivity and their cagemate before being tested. Water and food were available *ad libitum*. The study was approved by Texas A&M University's Animal Care and Use Committee (IACUC#2019-0219).

### Experimental procedure

During each trial, a pair of birds (consisting of a focal bird and non-focal bird; the initial designation of birds as the focal or non-focal bird was randomly assigned within each pair) was transported within its cage from the 'housing room' to an adjacent room ($2.2 \times 2.6$ m) that contained a sound attenuation chamber ($1.12 \times 0.67 \times 0.57$ m). The pair, within its cage, was placed within the middle of this chamber, which was visually and

acoustically isolated from the other captive birds in the 'housing room'. For 30 min, the birds' behavior was recorded using two video cameras (VIXIA HF R70; 60 frames/s; Cannon Inc., Ota City, Tokyo, Japan) positioned on each side of the chamber. After the 30 min, the experimenter (MRF) briefly removed the focal bird from his cage and intramuscularly injected 0.05 mL of a drug treatment or control into his breast. An experimenter monitored the birds (through remote videos) throughout their time in the chamber to ensure there were no signs of pain or adverse reactions from injection. The drug treatment consisted of a D2 agonist (PPHT; 1 mg/kg; Santa Cruz Biotechnology, Dallas, TX, USA) or a D2 antagonist (raclopride; 10 mg/kg; Santa Cruz Biotechnology, Dallas, TX, USA); the control consisted of 0.9% saline. Both drugs were dissolved in 0.9% saline. The doses were chosen to be high enough to insure they produced a behavioral effect and were the same as those used in previous avian studies (*Balthazart, Castagna & Ball, 1997*; *Zawilska, Derbiszewska & Nowak, 1996*); similar to these previous studies, all of the birds also received the same amounts of the drug treatment or control (the doses were calculated using 28 g as the average weight of the birds). Immediately following the injection, the focal bird was returned to his cage for another 30 min and their behavior was recorded. Once injected, the drugs were likely taken up into the brain rapidly (within 2 min in Sprague-Dawley rats (*Mukherjee et al., 2004*)). The non-focal bird remained within the chamber throughout the trial and did not receive any injections.

Each focal bird was tested in three separate trials in which he was administered either the D2 agonist, D2 antagonist, or saline (the order of the drug treatments and control was randomized across birds using a random number generator). Two days after the focal bird's trials were completed, the above experimental procedure was repeated except that the designation of the birds reversed: the previously non-focal bird in the pair was designated as the focal bird while the previously focal bird became the non-focal bird. Because the half-life of raclopride and PPHT are both estimated to be approximately 30 min (*Mukherjee et al., 2004*; *Köhler et al., 1985*), 2 days between trials ensured that the drugs were eliminated from the body between trials. All six trials of a given pair were conducted within 95 days of the birds' capture from the wild (mean ± SD: 31 ± 25 days) and between the hours of 08:00 and 13:00. Birds were released to their capture site at the conclusion of the study. Due to the short half-life of the treatment drugs, we did not anticipate that the birds would have adverse effects after release.

## Behavioral analysis

The behavior of the focal birds was scored from the video recordings. During a 10-min period preceding the injection ("pre-period") as well as a 10-min period following the injection ("post-period"), we recorded the amount of time the focal birds spent biting inanimate non-food objects (including the cage, perches, or their own leg bands) with their bill (henceforth 'biting'). In some species, biting is a coping behavior that can reduce stress associated with physical restraint, food restriction, or unfavorable environmental conditions (*Cabezas et al., 2013*; *Fossat et al., 2015*; *Hanson et al., 2020*). During the pre- and post-periods, we also scored the amount of time the focal birds engaged in intraspecific aggression (pecking or biting the non-focal bird), preening (cleaning feathers with the

bill or feet), bill wiping (rubbing bill on perch), feather ruffling (fluffing up and shaking feathers) and general movement (hopping or flying around the cage). Behaviors were scored by recording the frames at which the behaviors began and ended. We calculated the total amount of time the birds engaged in each behavior during the pre- and post-period for each treatment or control. One experimenter (MRF) scored all of the videos frame-by-frame using QuickTime (version 7; 60 frames/s; Apple Inc., Los Altos, CA, USA) and was blinded to the treatment.

## Statistical analysis

All statistical analyses were performed in R, Version 3.6.2 (*R Core Team, 2019*) and all models were performed using the package "lme4" (version 1.1-27.1; *Bates et al., 2015*). For each trial, we calculated a behavioral change score by subtracting the amount of time the birds engaged in each behavior during the pre-period from the amount of time the birds engaged in each behavior during the post-period. Because there were many trials in which biting, preening, and aggression did not occur, we adopted a two-step approach to account for this zero-inflated data. In the first step ("binomial model"), we performed a binomial mixed effect model with the response variable as whether or not the birds performed any biting, preening, or aggression behavior after injection; the treatment/control, trial order, change in weight during captivity, and change in general movement behavior were fixed effects in this model. In the second step ("count model"), we performed a linear mixed effect model using only trials where the biting, preening, or aggression behavior occurred after injection; we used the treatment/control, trial order, change in weight during captivity, and change in general movement behavior as fixed effects in this model. Because there were only two trials in which the birds exhibited biting after the antagonist injection, it was not possible to perform a statistical analysis on this group and it was therefore removed from the biting model in the second step ("count model") of this analysis. General movement was characterized by a single linear mixed effect model with change in general movement as the response variable and treatment/control, trial order, and change in weight during captivity as fixed effects. We included individual bird identity as a repeated measure in all models. For any model in which the treatment/control type was significant, we then performed pairwise comparisons to evaluate differences among the treatment/control using the R package 'emmeans' (version 1.7.0; *Searle, Speed & Milliken, 1980*) and used a Bonferroni correction to evaluate statistical significance. Feather ruffling and bill wiping were both rare behaviors, occurring on average less than 2% and 1% of the time in each trial respectively and therefore were not included in the analysis. Raw data is available in the Supplemental Information.

## RESULTS

The treatment/control was the only significant predictor of whether biting occurred or not ($n$ = 20, F = 4.31, $p$ = 0.001; Table 1; Fig. 1). Biting occurred in 19 out of 20 trials after administering the agonist, 2 out of 20 trials after the antagonist, and 4 out of 20 trials after saline. The treatment/control was also the only significant predictor of the amount of time

**Table 1 Results from the two-part models for biting, preening and aggression and the single model for general movement.**

| Response variable | Independent variables | Numerator df, denominator df | F-value (p-value) |
|---|---|---|---|
| Biting (Binomial model) | | | |
| | Treatment/control | 2, 53.0 | 4.31 (0.001)* |
| | Trial Order | 2, 53.0 | 1.24 (0.21) |
| | General movement | 1, 53.0 | 0.41 (0.77) |
| | Weight change | 1, 53.0 | 2.04 (0.15) |
| Biting (Count model) | | | |
| | Treatment/control | 1, 4.49 | 33.04 (0.003)* |
| | Trial Order | 2, 4.83 | 1.76 (0.26) |
| | General movement | 1, 9.81 | 3.03 (0.11) |
| | Weight change | 1, 18.99 | 1.85 (0.18) |
| Preening (Binomial model) | | | |
| | Treatment/control | 2, 53.0 | 0.48 (0.71) |
| | General movement | 1, 53.0 | 1.06 (0.49) |
| | Trial order | 2, 53.0 | 0.58 (0.35) |
| | Weight change | 1, 53.0 | 2.73 (0.09) |
| Preening (Count model) | | | |
| | Treatment/control | 2, 14.0 | 0.40 (0.67) |
| | General movement | 1, 14.0 | 6.25 (0.02)* |
| | Trial order | 2, 14.0 | 1.61 (0.23) |
| | Weight change | 1, 14.0 | 3.10 (0.09) |
| Aggression (Binomial model) | | | |
| | Treatment/control | 2, 53.0 | 2.06 (0.13) |
| | General movement | 1, 53.0 | 0.07 (0.87) |
| | Trial order | 2, 53.0 | 0.24 (0.85) |
| | Weight change | 1, 53.0 | 0.73 (0.38) |
| Aggression (Count model) | | | |
| | Treatment/control | 2, 2.91 | 1.32 (0.38) |
| | General movement | 1, 11.36 | 1.77 (0.20) |
| | Trial order | 2, 3.73 | 1.05 (0.43) |
| | Weight change | 1, 14.23 | 0.02 (0.88) |
| General movement | | | |
| | Treatment/control | 2, 36.0 | 1.53 (0.22) |
| | Trial order | 2, 36.0 | 1.91 (0.16) |
| | Weight change | 1, 18.0 | 2.21 (0.15) |

**Note:**
Statistical significance is indicated with an asterisk.

spent biting (assuming time biting was above zero). The birds spent more time biting when administered the D2 agonist compared to the control ($n = 19$, t = 4.84, $p = 0.002$; Table 2). No other variable predicted the presence or amount of biting (Table 1).

The treatment/control had no effect on either the presence or amount of preening, aggression, or general movement ($n = 20$, F = 0.40, $p = 0.67$; $n = 20$, F = 1.32, $p = 0.38$;

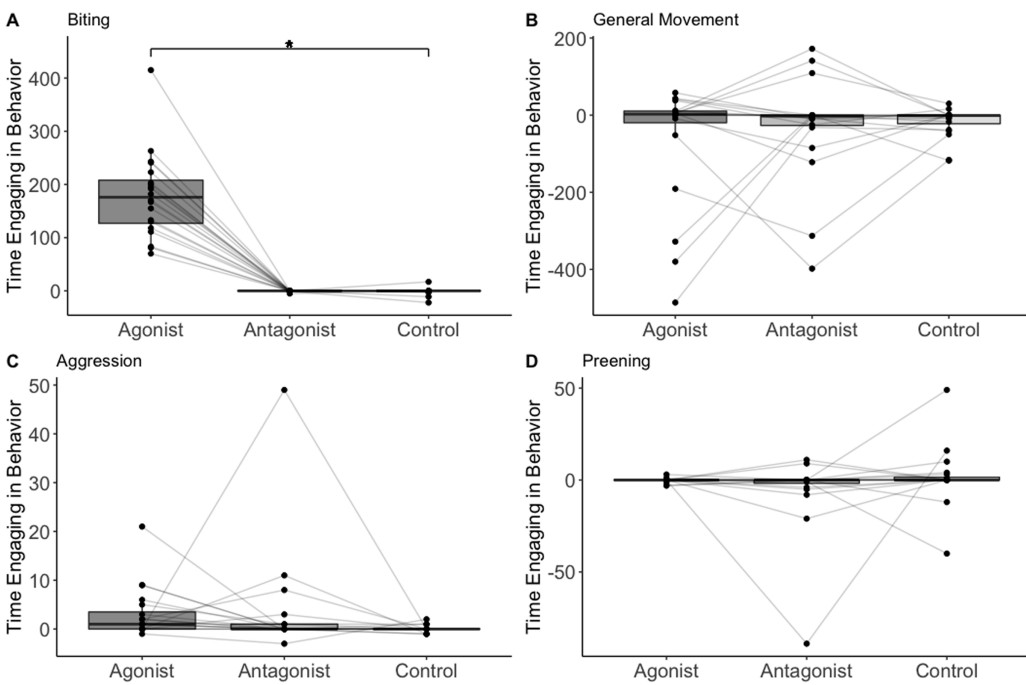

**Figure 1 The change in number of frames birds ($n$ = 20) spent on biting (A) general movement (B), aggression (C), and preening (D).** Data points representing the same individual are connected with lines. Asterisks indicate statistical significance.

**Table 2 Pairwise comparisons between treatment/control for the two-part biting model.**

|  | Numerator df, denominator df | t-ratio (*p*-value) |
|---|---|---|
| Comparisons (Biting Binomial model) |  |  |
| Control *vs* Agonist | 1, 1.62 | 3.21 (0.003)* |
| Control *vs* Antagonist | 1, 1.07 | 1.19 (0.69) |
| Agonist *vs* Antagonist | 1, 1.82 | 3.57 (0.001)* |
| Comparisons (Biting Count model) |  |  |
| Control *vs* Agonist | 1, 6.11 | 4.84 (0.002)* |
| Control *vs* Antagonist | -* | -* |
| Agonist *vs* Antagonist | -* | -* |

Note:
Statistical significance is indicated with an asterisk.
\* Because only two birds exhibited biting after the injection in the antagonist trials, it was not possible to perform a statistical analysis on comparisons involving the antagonist.

$n$ = 20, F = 153, $p$ = 0.22; Table 1). The birds were less likely to preen when they spent more time moving ($n$ = 20 F = 6.25, $p$ = 0.02; Table 1). No other variable predicted presence or amount of preening, aggression, or general movement (Table 1).

## DISCUSSION

We found possible support for the hypothesis that dopamine influences stress-related coping behavior in a wild-caught species. In particular, the house sparrows spent more time biting at inanimate objects when they were administered the D2 agonist. Biting at

inanimate objects may be a coping mechanism in some bird species (*Savory & Kostal, 1993*; *Nicol, 1987*; *Zarrindast, Hajian-Heydari & Hoseini-Nia, 1992*), but has not yet been demonstrated in passerines or any other wild species. The amount of time house sparrows spent biting did not decrease when they were administered the D2 antagonist, although this is not surprising as they rarely (only two birds) exhibited biting prior to the antagonist administration. Biting behavior also did not increase when administered a saline control, indicating that the behavior is unlikely to be a response to stress induced by handling and injection. We also did not find any associations between the biting behavior and weight loss in captivity; weight loss is often a sign of stress from captivity (*Parker Fischer, Wright-Lichter & Romero, 2018*) and this therefore suggests that stress associated with captivity did not influence the biting behavior. Furthermore, increases in dopamine receptor activation are often associated with increases in general movement (*Beninger, 1983*). However, even after controlling for general movement, we still found that the house sparrows spent more time biting at inanimate objects when administered the D2 agonist.

In rodent models as well as poultry, environmental stressors elevate dopamine levels in the mesolimbic system (*Cabib & Puglisi-Allegra, 1996*; *Cheng, Singleton & Muir, 2003*). We therefore speculate that the activation of D2 dopamine receptors by the agonist could induce neurochemical changes consistent with stress in the house sparrows and the birds behaviorally coped with this perceived stress by biting at inanimate objects. Coping behaviors are de-arousal mechanisms that work to restore homeostasis after a disturbance (*Savory & Kostal, 2006*). Domesticated chickens (*Gallus gallus*) bite at inanimate objects when they experience stressful conditions such as high densities and low temperatures (*Spinu, Benveneste & Degen, 2003*) and those that engage in more pecking at inanimate objects had lower corticosterone levels (*Kostal, Savory & Hughes, 1992*). Rodents that bite at inanimate objects during stressful situations such as restraint and introduction into a new environment also have lower levels of cortisol and corticosterone, respectively (*Sato et al., 2010*; *Hennessy & Foy, 1987*). The behavior observed in this experiment may also act as a coping behavior, however we did not measure physiological measures of stress such as corticosterone or heart rate. Therefore, future studies are needed to determine whether the observed biting behavior also impacts stress levels in wild songbirds.

While it is possible that the house sparrows exhibited biting behavior to cope with stress, it is also possible that the biting behavior serves no purpose or even has negative effects. Repeated biting at inanimate objects is not a behavior typically observed in wild sparrows, making it a possible stereotypy (defined as a repetitive movement that serves no apparent purpose; *Powell et al., 1999*). Stereotypic behaviors can have positive effects; for example, voles performing stereotypy in captivity have higher rates of survival and reproduction, although the mechanism is unknown (*Schønecker, 2009*). However, stereotypy can also have neutral or negative effects (*Mason & Latham, 2004*). For example, tufted capuchins (*Cebus apella*) that exhibited stereotypic head twirling behaviors exhibited a more negative affective state compared to conspecifics that did not exhibit these stereotypic behaviors (*Pomerantz et al., 2012*). Stereotypic feather plucking in African gray parrots (*Psittacus erithacus*) is also associated with higher levels of fecal

corticosterone suggesting they are experiencing chronic stress that is not extinguished by the stereotypy (*Costa et al., 2016*; *Owen & Lane, 2006*). A study in quail that (*Balthazart, Castagna & Ball, 1997*) used the same D2 agonist at the same concentration as our study; the quail exhibited extremely high rates of pecking behavior that were interpreted as a stereotypy and their pecking behavior may be analogous to the biting observed in this study. Therefore, the dopamine agonist in this study may be producing abnormal conditions in the brain leading to stereotyped behaviors that do not aid in coping with stress.

It is unlikely that the house sparrows exhibited increased biting because they were hungry. Although the birds did not have access to food during any of the trials, they only increased their biting after administration of the D2 agonist but not the D2 antagonist or saline. Rodents and Japanese quail also increased their biting behavior after they were given a D2 agonist, but they did not increase their food consumption even though food was available (*Nicol, 1987*; *Beninger, 1983*). This suggests that biting induced by D2 receptor activation is not related to an increased motivation to feed.

We also did not find support for the possibility that the birds increased biting because they became more aggressive. There was no relationship between biting inanimate objects and intraspecific aggression. Furthermore, the house sparrows exhibited very low levels of aggression across all trials. Similarly, manipulating dopamine receptor activation in group-housed chickens did not influence aggression towards conspecifics (*Dennis, Muir & Cheng, 2006*). These results indicate that the D2 receptors are not involved in aggression among familiar conspecifics, but further experiments could examine if this is also the case among unfamiliar conspecifics that are more likely to engage in aggression (*Hegner & Wingfield, 1987*).

We found that the D2 antagonist did not influence biting or any other behaviors in house sparrows. Because the birds rarely exhibited biting before the D2 antagonist treatment, there was little opportunity for the behavior to become even less frequent. In fact, only two birds exhibited biting before or after the D2 antagonist injection. However, the treatment may have effects when the behavior is already present. When chickens are chronically stressed, D2 antagonists reduced the amount of time they spent biting at inanimate objects (*Savory & Kostal, 1993*). Furthermore, a different dopamine antagonist administered to rats only had an impact on behavior after multiple treatments (*Ikemoto & Panksepp, 1996*). Additional studies could examine whether D2 antagonists likewise alter coping behavior in songbirds when their stress levels are high.

## CONCLUSIONS

We found that an increase in D2 dopamine receptor activation in house sparrows leads to increased biting of inanimate objects. Because biting behavior may be a coping behavior to reduce stress (*Savory & Kostal, 1993*; *Nicol, 1987*), our results could indicate that dopamine is involved in songbirds' stress response. However, the observed behavior may instead be an expression of stereotypic behavior. D2 dopamine receptor activation alters biting behavior in domesticated birds (*Dennis, Muir & Cheng, 2006*; *Savory & Kostal, 1993*; *Zarrindast, Hajian-Heydari & Hoseini-Nia, 1992*), indicating that the dopamine

pathways between these domestic species and the wild songbird in this study are likely similar. While many studies have examined the influence of hormones on the stress response (*Schoech, Rensel & Heiss, 2011*; *Creel et al., 2013*), our understanding of how neurotransmitters impact stress in wild species is still poor (*Lattin et al., 2019*; *Trainor, 2011*). To further understand this question, research on the effect of dopaminergic drugs on birds that were socially or physiologically stressed compared to unstressed controls would be valuable. It would also be interesting to look at the effects of different dosages of these drugs, to see if the behavioral effects were specific to high dosages such as reported by *Balthazart, Castagna & Ball (1997)*. Finally, we only used male birds in this experiment and further work is needed to determine if sex has an influence on dopamine's effect on the stress response. Wild species are increasingly exposed to stress due to human disturbances (*Birnie-Gauvin et al., 2016*). Our current understanding of how animals perceive and cope with stress mostly comes from domestic animals, which may respond to stress in different ways than their wild counterparts. Therefore a greater understanding of how wild organisms respond to stress will be crucial to evaluate how these organisms will manage stressors in their environments.

## ACKNOWLEDGEMENTS

We would like to thank Margaret Guy for her assistance in performing the experiment. We would also like to thank Drs. Jeffery Tomberlin, Sarah Hamer, and Gil Rosenthal for feedback on an earlier version of this manuscript.

### Funding

Jessica L. Yorzinski was supported by the National Science Foundation (BCS #1926327), the College of Agriculture and Life Sciences at Texas A&M University and Texas A&M AgriLife Research. Melanie R. Florkowski was supported by the College of Agriculture and Life Sciences and the College of Science at Texas A&M University as well as Texas A&M AgriLife Research and Texas A&M University's Office of Graduate and Professional Students. Texas A&M University's Department of Ecology and Conservation Biology provided funding for open access costs. The funders had no role in study design, data collection and analysis, decision to publish, or preparation of the manuscript.

### Grant Disclosures

The following grant information was disclosed by the authors:
National Science Foundation: 1926327.
College of Agriculture and Life Sciences at Texas A&M University.
Texas A & M AgriLife Research.
College of Agriculture and Life Sciences.
Texas A&M University's Department of Ecology and Conservation Biology.

## Competing Interests

The authors declare that they have no competing interests.

## Author Contributions

- Melanie R. Florkowski conceived and designed the experiments, performed the experiments, analyzed the data, prepared figures and/or tables, authored or reviewed drafts of the article, and approved the final draft.
- Jessica L. Yorzinski conceived and designed the experiments, authored or reviewed drafts of the article, and approved the final draft.

## Animal Ethics

The following information was supplied relating to ethical approvals (*i.e.*, approving body and any reference numbers):

The Texas A&M University's Animal Care and Use Committee approved the study (IACUC# 2019-0219).

## Data Availability

The raw data is available in the Supplemental File.

## Supplemental Information

Supplemental information for this article can be found online at http://dx.doi.org/10.7717/peerj.13520#supplemental-information.

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
