# Peer review of "Dopamine receptor activation elicits a possible stress-related coping behavior in a wild-caught songbird"

_PeerJ, doi:10.7717/peerj.13520_

## Round 0.1 · original submission · Major Revisions

We have managed to obtain two reviews on your manuscript. Both reviewers found that the manuscript was clearly written and interesting. However they both underlined important points that will deserve further revisions, as you will find in their reports below. In particular, reviewer 1 doubted the general interpretation of your experiment/results. Also, reviewer 2 questioned the validity of the statistical analyses and asked for more details concerning the models being used and for a clearer presentation of data.

I have also myself assessed your manuscript and found a few important points that need to be better justified for a clearer understanding of the experiment.

First, like reviewer 1, I wondered if biting at objects is a stress coping behavior in house sparrows? How is this supported in the wild and in passerines for example? I thus also agree with the reviewer that the language connecting biting to stress-coping behavior should be greatly toned down throughout the manuscript and that alternative explanations should be presented more clearly.

Also, like reviewer 2, I think that a clearer description of the statistical analyses is warranted and I wondered how was the zero-inflated data taken into account?

Finally, I wondered if this kind of dose was used previously in passerines or in wild birds as you only cited references related to domestic (and much larger) birds in the methods, please clarify.

Other minor points:
L98: why only males?
L150: it is thus a linear mixed model

·

Basic reporting

The manuscript by Florkowski and Yorzinski entitled “Dopamine receptor activation elicits a possible stress-related coping behavior in a songbird” assesses the behavioral effects of peripheral administration of dopamine modulators in wild-caught house sparrows. The authors find that a D2 agonist increases “inanimate object biting”, when compared to an antagonist or saline. No other behaviors changed significantly.

Overall, the manuscript is clearly written and structured, experiments were well designed, and data analysis is solid. Raw data were shared.

Some minor comments:
a) (lines 23-24, 85-86) Please provide rationale for the importance of studying stress responses in this species. The authors state that “few studies have examined[…]” and “there have so far been few studies investigating[…]”, but fail to clearly state the importance of this investigation. In the case of the house sparrow, it could be interesting to make connections between stress responses and their incredible invasiveness (and success) into ultra-urbanized environments. Could their stress responses be adaptive and underlie this? You might find Beaugeard et al., 2018 (Ecology and Evolution) helpful for this.

b) Figures are mostly clear and well-formatted. Minor stylistic comments:
- Figure 1 would benefit from a legend for the barplot colors. Additionally, plotting individual datapoints (overlaid on top of bars) would give a better sense of variability than only means and SEMs.
- Figure 2 would benefit from larger marker sizes as well as a color scheme to match the one in figure 1

Experimental design

Experiments were well designed and executed. Methods are well described. Data are well analyzed and reported. Statistical tests are appropriate and well-controlled. Overall, authors did a great job here.

Minor comments:
a) (line 117) Purchase source for PPHT is missing.
b) (line 134) This could be a good spot to present justification (if it exists) that releasing these birds back into the wild after treatments and captivity is not expected to produce any deficits in fitness.
c) (line 136) Did you use any specific software for helping score videos?
d) Weight and body condition could vary significantly in wild-caught birds. Did you record birds’ weights and assess overall body condition? If so, body weight could be used as a random variable in your LMs. Did you administer any anti-parasitic treatment upon lab admittance?

Validity of the findings

The data upon which the conclusions are based are robust and well-controlled. However, my main concern with the manuscript is that the way the data are interpreted does not fully align with the conclusions, such that alternative interpretations cannot be ruled out.
In my opinion, the interpretation of “inanimate object biting” as a stress coping mechanism is somewhat generous and not fully supported by the presented data. The main caveat of this report is that it lacks any direct measures of stress – e.g., chronic stress markers (weight change, feather loss, aggression), or physiological markers such as plasma corticosterone levels.
To be clear, the authors do their due diligence to present relevant literature in support of the possible link between D2 receptor activation and stress-coping behaviors (including biting). I agree this is a possible explanation. However, the authors acknowledge other possible interpretations for this drug-induced behavior (lines 202-210). It is not clear to me why some of these (particularly stereotypy) are not just as likely as stress-coping.
This issue could be mitigated by “toning down” the language that directly connects biting behavior to stress-coping throughout the manuscript (including in the title).

Reviewer 2 ·

Basic reporting

This manuscript by Florkowski and Yorzinski investigates the role of the dopaminergic system in coping behaviors in a wild songbird. I found the manuscript to be well-written, with a sound experimental design that allows the authors to answer their research question. The Introduction and Discussion provide good context for the present experiment, and the data strongly suggest that D2 receptor activation increases coping-like behaviors in the absence of major stressors in this species. I had concerns with the statistics and presentation of the data that should be addressed prior to publication. I enjoyed reading the manuscript and thinking about the data, and look forward to the authors’ revisions.

Experimental design

I thought that the research is original and within the aims and scope, that the research question was well defined, relevant & meaningful, that the study design was rigorous. See the "Validity of the findings" section below for some methods details that should be included.

Validity of the findings

Major comments:
• I didn’t find the statistical analysis to be appropriate for the authors’ data. The large error bars indicate a wide spread of behavior, and figure 2 shows that this might be due to many birds not performing a behavior (i.e., 0-inflated data), as well as the raw data. There was also a possible issue with reporting of pre- and post- treatment behavior. Based on this, I have the following comments:
o Figure 1 shows that there are possible differences from the pre to post period in selective treatment groups, but the statistics don’t detect this. This is either because these aren’t real differences (i.e., large variances are due to a couple of individuals that did a lot of behavior), or because of the 0-inflated data (i.e., a spread of individuals who did the behavior, but the error is stretched wide because many birds also didn’t do the behavior at all).
o I am not totally clear on the models that were run from the authors description in the methods. Could they help to clarify? Table 2 helps clear up some, but there should either be a description in the methods or more detail in the results about the “negative” findings.
o I am not clear on whether the models were nested appropriately. For example, each set of pre- and post- treatment behavior are separate, so it would be inappropriate to run the pre- and post- difference as a main effect. Rather, there are pairs of pre- and post- for each treatment (i.e., three treatments per individual, and one pair of pre- post- specific to each treatment). I don’t believe the random effect controls for this nested design.
 I could see a few ways to address this, including running separate models for the pre- to post- comparisons (and the individual, as well as the pair, would have to be taken into account), or subtracting pre behaviors from post behaviors to get a difference score (or maybe even a normalized Z score to control for the variance between animals in the pre-period for each treatment).
o I don’t have a specific preference for the way the authors handle the 0-inflated data, but their approach should answer whether the number of animals that do 0 behaviors obscure treatment effects for the proportion of animals that engage in the behavior. Possible approaches include zero-inflated poisson models or negative binomial models, or conducting the current analysis with and without animals with zero behaviors and reporting the results both ways.
o The authors state that there is no relationship between biting and movement, but this just isn’t true. Figure 2 shows that birds either spend their time moving, or biting depending on the treatment (apart from the many animals that engaged in neither behavior), and do not do intermediate levels of both. This is very interesting and relevant to the authors’ research question.
 As for how to put some kind of statistical test to this relationship, I leave that to the authors to decide what is most appropriate given the data (and my comment about possible analyses above).

• Given that the data are repeated measures, the figure presentations should reflect this.
o While readers can download the raw data, I think the authors should plot individual data points on their plots in figure 1 (also see below for comments about visualizing repeated measures data)
o Bar plots with error bars do not accurately reflect the nature of the data. One approach would be to connect individual data points with lines, laid over bar plots (or next to them). This could be done in many ways depending on how the authors wish to present the data. As examples, data points could be connected between all 6 bars (i.e., keeping the organization of Figure 1), or the authors could rearrange, or only show repeated measures between post- treatment groups, or plot the difference scores between post- and pre- and connect data points between these.


Minor comments:
• General methods question: Were birds in the same pair given the same treatment each time, or different treatments? If different treatments, could this have any effect on the other partner?
o Lines 142-145: Correct these sentences to say that all behaviors were scored in both the pre and post periods.
o Please add a legend to figure 1 for the treatment group colors.
o Many of the results for the variables the authors included in their models are not presented in the results text. An easy fix would be to say that “no other independent variables significantly predicted time spent engaging in behavior (Table 2).” But see concerns about appropriateness of stats and data visualization.
o Lines 248-251: I think it would emphasize the rationale of the study to add in a point the authors made in the introduction down here, that there’s a need to test these questions in wild species because domestic species appear to have different physiological responses (and possibly to add in a short discussion of why this might be…totally optional, just a suggestion).

---

## Round 0.2 · Minor Revisions

We obtained additional reviews from both previous reviewers on your study. They both felt that the new version was greatly improved and also suggested a few additional minor points that need to integrated in the next version of the manuscript.

·

Basic reporting

The manuscript by Florkowski and Yorzinski entitled “Dopamine receptor activation elicits a possible stress-related coping behavior in a songbird” assesses the behavioral effects of peripheral administration of dopamine modulators in wild-caught house sparrows. The authors find that a D2 agonist increases “inanimate object biting”, when compared to an antagonist or saline. No other behaviors changed significantly. The authors suggest that inanimate object biting could be a stress-coping behavior; therefore this finding supports that D2 activation induces stress-coping behavior.
Overall, the manuscript is clearly written and structured, experiments were well designed, and data analysis is solid. Raw data were shared.
The authors addressed my previous concerns satisfactorily. I only have some further minor comments to improve readability and clarity.
a) (line 23) Replace “which are” with “i.e.”
b) (line 24-25) If you have space in the abstract, summarize the excellent rationale you provided about studying stress in house sparrows (lines 90-91)
c) (line 85) Italicize the species name
d) (lines 85-87) Relocate the Lattin 2019 citation to immediately follow the fragment “A study in wild-caught house sparrows (Passer domesticus)”
e) (line 86) Replace “dopamine” with “D2”
f) (lines 90-91) Excellent!
g) (lines 192-193) Add information about on the ratio of how many (x out of N) animals performed biting after treatment
h) (line 273) If D2 agonists increase stress-coping behaviors which in turn decrease stress, then the logic here would follow that D2 agonists would decrease stress-coping behaviors, therefore increasing stress

Experimental design

Experiments were well designed and executed. Methods are well described. Data are well analyzed and reported. Statistical tests are appropriate and well-controlled. All my minor concerns were addressed. I also liked the addition of the zero-inflation data correction.

Validity of the findings

My major concern was addressed by the authors. The language about linking inanimate object biting to stress-coping was toned down, the caveats of the study are more explicitly stated, and the alternative interpretations are appropriately acknowledged.

Reviewer 2 ·

Basic reporting

• I appreciate the revisions the authors made throughout the manuscript, and I only have minor comments.
• I think the title could benefit from adding “wild” or “wild-caught” as a descriptor of songbird; it would better describe an important aspect of the authors’ research question.
• I agree with the addition of ecological context for the species, but would suggest replacing the term “urban invasive” with a more accurate descriptor (with respect to the research question at hand). On line 96, something like “success in non-native urban environments”, and on line 100, something like “urban specialist”.
• Line 166: Typo: change “we” to “were”
• Line 184: Typo: add “a” to “as repeated measure”
• Lines 239-240: It was unclear from the writing what the authors mean by stereotypic behaviors compensating for the unnatural environment.

Experimental design

• Lines 167-170: clarification, mismatch between wording in data – wording suggests the authors did pre minus post.
• See below for validity of the findings, but briefly, the authors have addressed the concerns I had related to the statistical analysis.

Validity of the findings

• I thank the authors for their thorough responses to my feedback. I agree with them that the new analyses have not altered the major findings of the manuscript and that the statistics and data visualization are appropriate and well-executed.
• With respect to Reviewer 1’s comment from the previous review, please make a change on Line 26 in the abstract…something like changing stress-related coping behavior to “coping-like behavior”.
• Also with respect to Reviewer 1’s comment, I noted that the dose of PPHT (1 mg/kg) used in the present study, also used in the Balthazart 1997 study, led Balthazart et al. to a pretty vivid description of pecking behavior. I agree their description doesn’t mean the behavior is definitively stereotypy rather than something more akin to coping, but the authors can more directly address this in the discussion in their newly added paragraph about alternative explanations beginning on line 234 (that the same dose in two different bird species elicits a very strong and unique behavior not commonly observed in other conditions, and that the interpretation in the quail study is stereotypy).
• Line 207: the authors could consider making the following addition “…not yet been demonstrated in any wild species, or in passerines”.
• Lines 220-223: also with respect to Reviewer 1’s comment about alternative explanations, because the authors did not find a link between weight loss in captivity and biting induced by the agonist, is it equally likely that signaling at D2 leads to biting behavior independent of stress? This wouldn’t be mutually exclusive with the authors’ suggestion that stress may result in increased biting behavior.

---

## Round 0.3 · accepted · Accept

I am satisfied with the final revisions made to the manuscript.